

# Comparison of the optical properties of pure and transported anthropogenic dusts measured by ground-based Lidar

Zhijuan Zhang[1], Bin Chen[1,*], Jianping Huang[1], Jingjing Liu[2], Jianrong Bi[1], Tian Zhou[1], Zhongwei Huang[1]

[1]Key Laboratory for Semi-Arid Climate Change of the Ministry of Education, College of Atmospheric Sciences, Lanzhou University, Lanzhou, 730000, China

[2]School of Mechanical and Instrument Engineering, Xi'an University of Technology, Xi'an 710048, China

[*]Corresponding author:chenbin@lzu.edu.cn



**Abstract**

In this study, the optical properties of pure dust (PDU) and transported
anthropogenic dust (TDU) (also defined as polluted dust) are compared by using
ground-based Lidar data for the period from October 2009 to June 2013. The total
attenuated backscattering coefficient at 532 nm, the linear volume depolarization ratio
and the color ratio are derived from the L2S-SM-II dual-band polarization Lidar. We
found that the TDU has a spherical shape, a small linear volume depolarization ratio
and a large color ratio which representing its large particle sizes. The threshold value
delineating PDU and TDU was approximately 0.2, which is the same as the threshold
value used in the CALIPSO CAD algorithm. The histogram of the attenuated
backscattering coefficients and the color ratios of pure dust shows two peaks, but that
for the transported anthropogenic dust shows no significant peak and a nearly uniform
distribution. The ground-based Lidar results confirm that both the transported
anthropogenic dust and pure dust can be detected by air-borne or ground-based Lidar
measurements.






## 1 Introduction

Dust aerosols are one of the most important aerosol types in the troposphere and are an important source of atmospheric aerosols (Huang et al., 2014). Dust can impact the earth-atmosphere radiation budget by absorbing and scattering solar radiation as well as by emitting IR radiation (direct effect) (e.g., Sokolik and Toon, 1996; Li, 2004; Shi et al., 2005, Huang et al., 2009), altering the optical properties and lifetimes of clouds (indirect effect) (e.g., Sassen, 2002), increasing the evaporation of cloud droplets and further reducing the CWP (Cloud Water Path) by the means of warming clouds (semi-direct effect) (Huang et al., 2006b), all of which can eventually change the climate (Luo et al., 2000, Twomey et al., 1984; Huang et al., 2005, 2006a, 2006b;), especially in semi-arid regions in East Asia (Huang et al., 2010, 2014). Dust aerosols, or mineral dusts, have obvious heating or cooling effects that can change the atmospheric thermal circulations and dynamic conditions, making dust aerosols one of the important factors triggering global environmental problems. However, the existing atmospheric dust load cannot be explained by natural sources alone. The atmospheric dust load that originates from soils disturbed by human activities, such as land use practices, can be interpreted as "anthropogenic" dusts (Tegen and Fung, 1995). Anthropogenic dusts are those produced by human activities on disturbed soils, which are found mainly in croplands, pasturelands, and urbanized regions, and are a subset of the total dust load, which includes natural sources from desert regions (Huang et al., 2015).

Local anthropogenic dust aerosols associated with human activities, such as



agricultural and industrial activities, accounted for 25% of the total dust burden in the
atmosphere (Huang et al., 2015). These anthropogenic dusts can increase dust loading,
which, in turn, affects radiative forcing (Tegen and Fung, 1995). Huang et al. (2015)
found that local anthropogenic dust aerosols from human activities, such as
agriculture, industrial activity, transportation, and overgrazing, account for
approximately 25% of the global continental dust load. Of these anthropogenic dust
aerosols, more than 53% come from semi-arid and semi-wet regions (Guan et al.,
2016). The annual mean anthropogenic dust column burden values range from a 0.42
g m$^{-2}$ maximum in India to a 0.12 g m$^{-2}$ minimum in North America. Previous works
have also explored the global relationship between anthropogenic dusts and
population over semi-arid regions. The results showed that the relationship between
anthropogenic dusts and population is more obvious for croplands than for other land
cover types (crop mosaics, grassland, and urbanized regions). The production of
anthropogenic dust increases as the population density grows to more than 90 persons
km$^{-2}$. The most significant relationship between anthropogenic dust and population
occurred in an Indian semi-arid region that had a high portion of croplands, and the
peak anthropogenic dust probability appeared at a 220 persons km$^{-2}$ population
density and a 60 person km$^{-2}$ population change.
In earlier publications (Tegen and Fung, 1995; Huang et al., 2015),
anthropogenic dusts were described at the portion of mineral dust that is primarily
produced by various human activities on disturbed soils (e.g., agricultural practices,
industrial activity, transportation, desertification and deforestation). East Asia has the





highest concentration of anthropogenic aerosols in the world (Sugimoto et al., 2015a).
Additionally, East Asia is a unique region wherein mineral dust (Asian dust) sources
are located near urban and industrial areas. During transportation, dust often mixes
with anthropogenic aerosols (Takemura et al., 2002) and induces new environmental
and climatic problems (Su et al., 2008). In this paper, we attempt to study this kind of
transported anthropogenic dust (TDU), which is mainly dominated by dust and could
be mixed with other anthropogenic aerosol types. Although there are some
quantitative assessments about the anthropogenic dust, the accuracies of these results
are still unknown due to the limited data and preliminary detection methods. In
Huang's method (Huang et al, 2015), approximately 9.6% of the anthropogenic dust is
misclassified as natural dust, and 8.7% of the natural dust is misclassified as
anthropogenic dust within the PBL (planetary boundary layer).
Lidar, an advanced active remote sensing instrument with high spatial and
temporal resolutions and high accuracy detection abilities in the lower altitudes, has
become an important technology for detecting the spatial and temporal distributions
of the aerosol physical properties (Zhou et al, 2013). Hua et al. (2005a, 2005b, 2005c,
2005d, 2007) used ultraviolet Rayleigh–Mie Lidar and Raman Lidar for temperature
profiling of the troposphere. Chen et al. (2010) and Liu et al. (2011) used the
satellite-based Lidar CALIOP (Cloud-Aerosol Lidar with Orthogonal Polarization) to
detect the dust layers with fewer misclassifications. However, when detecting surface
dusts, ground-based Lidar has an obvious advantage over the satellite-based Lidar. In
this study, the ground-based Lidar measurements are used to validate the thresholds



used in the CALIPSO CAD algorithm (Liu et al., 2005). The total attenuated
backscattering coefficient at 532 nm, the linear volume depolarization ratio and the
color ratio are derived from the L2S-SM-II dual-band polarization Lidar developed by
the NIES (National Institute for Environment Studies) and provided at the Semi-Arid
Climate and Environment Observatory of Lanzhou University (SACOL).
The paper is arranged as follows. The details of the datasets used are given in
section 2. In section 3, the inversion and detection method used in this study is
introduced. Examples of distinguishing pure dust and transported anthropogenic dust
using multiple measurements are presented in section 4. A comparison of the optical
properties of two dust cases is presented in section 5. The conclusion and discussion
are presented in section 6.
**2 Data**
**2.1 Surface station data**
The global surface weather data set from the China Meteorological
Administration State Information Center was used in this study. This data set is based
on the global surface monthly data and real-time data, which are then decoded and
normalized. The time period of the data set spans from January 1, 1980 to June 1,
2015. There are 65 elements in every record of the data set, and the types of variables
are set as characters. The data set is strictly quality controlled. Here, we analyze the
weather phenomena from October 2009 to June 2013.
**2.2 Ground-based Lidar data**
The Semi-Arid Climate and Environment Observatory of Lanzhou University



(SACOL) (Huang et al., 2008, Guan et al., 2009, Wang et al., 2010, Huang et al., 2010,
Bi et al., 2010, Liu et al., 2011), built in 2006, is situated on the Loess Plateau
(35.946 °N, 104.137 °E) at approximately 1965.8 m above sea level. The topography
around the site is characterized by the Loess Plateau and consists of plains, ridges and
mounds, etc. The dominant species within the immediate area of the study site are
Stipa bungeana as well as Artemisia frigida and Leymus secalinus. SACOL is
approximately 48 km away from the center of Lanzhou. The terrain where the
measurements are made is flat and covered with short grasses. The reason that the site
was built on the mountain top is as follows: the environment of the mountain top is
almost completely natural and is rarely affected by human activity and the climate at
the site can represent that of the surrounding hundreds of kilometers. Thus, by
building at the top of the mountain, the influences of houses and other human
activities are avoided. The L2S-SM-II dual-band depolarization Lidar are operated at
SACOL and began observing aerosols and clouds in October 2009.

Fig. 2 shows the structure of the L2S-SM-II dual-band depolarization Lidar at

SACOL, which is a two-wavelength polarization-sensitive backscatter Lidar. The
NIES's vertically resolved aerosol and cloud measurements will enable new insights
into the roles of aerosols and clouds in the Earth's climate system. This Lidar system
consists of three parts: the laser source, signal receipt-system and data recording
device. The laser source is a flash lamp pumped Nd:YAG laser device. Two laser
beams (with wavelengths of 532 nm and 1064 nm) are shot into the atmosphere to
calibrate for the beam expanding, and the return signal is received by the Cassegrain



telescope with a diameter of 20 cm. The perpendicular and parallel components of the
532 nm backscatter signal are received by two detectors. Thus, we can derive
polarization information. Using the relationship of the delay time and the height at
which the light is scattered, the power of the return signal and the concentrations of
atmospheric aerosols are known. Therefore, the vertical profile of the optical
properties of aerosols can be derived (Zhou et al, 2013). The vertical resolution of the
Lidar structure is 6 m and can reach a height of 18 km above the ground. The time
resolution of the Lidar system is 15 min. For our study, we choose measurements
taken over a continuous period from October 2009 to June 2013.
**3 Retrieval and detection methods**
Lidar signals, such as the total attenuated backscattering coefficient at 532 nm,
linear volume depolarization ratio, and color ratio, reflect the physical and optical
properties of aerosols and clouds. There are a number of effective methods for
deriving particulate extinction and backscatter coefficients from calibrated,
range-corrected Lidar signals. Among these, the most widely used are the Klett
method (Klett, 1985), the Fernald method (Fernald, 1984) and the so-called linear
iterative method first introduced in the late 1960s (Elterman, 1966) that was
subsequently extensively used by Platt (Platt, 1973; Platt et al., 1998). The Fernald
algorithm was originally developed within the context of single scattering. In later
years, both algorithms were adapted for use in multiple scattering analyses via a
correction factor of the range-resolved extinction coefficients (e.g., as in (Young,





1995)). In our study, we adapt the Fernald method.

Generally, clouds are seen to have larger backscatter coefficients and higher

color ratios (~ 1) than aerosols. The exceptions to this general rule are desert aerosols
and maritime aerosols under high relative humidity conditions, both of which then
exhibit relatively large color ratios. These scattering features can be used to
distinguish aerosols from clouds. Additionally, the linear volume depolarization ratio
is a useful indicator for identifying irregular particles and provides a means of
discriminating ice clouds from water clouds and identifying dust aerosols. An
attenuated backscattering coefficient is vital in many aspects. Accurate aerosol and
cloud heights and the retrieval of extinction coefficient profiles are all derived from
the total backscatter measurements. Winker et al (2006) compared the sensitivities of
the 532 nm and 1064 nm channels. The APD detector used in the 1064 nm channel
has much higher dark noise than the PMT detectors used in the 532 nm channels. The
sensitivity of the 1064 nm channel is limited in most situations by the detector dark
current, so the sensitivity shows much smaller variations between days and nights and
over varying altitudes than the 532 nm channel. For this reason, the attenuated
backscattering coefficient at 532 nm is one of the best indicators for discriminating
aerosols and clouds.

The linear volume depolarization ratio is defined as the perpendicular

components of the 532 nm attenuated backscatter coefficient over the parallel
components of the same coefficient. The expression for this is as follows:

$$\delta(r) = \beta_{532,\perp}(r) / \beta_{532,\parallel}(r) \quad (1)$$





The sphericity of a particle is represented by its linear volume depolarization
ratio, such that a value near 0 indicates that the particle is nearly spherical, while a
large value indicates that the particle is aspherical. The linear volume depolarization
of ice crystals is typically in the range of 30%-50% but depends on the crystal shape
and aspect ratio. Lower values can be seen when horizontally oriented particles are
present (Sassen and Benson, 2001). In contrast, the backscattering from spherical
water droplets preserves the polarization of the incident light, so the value of the
linear volume depolarization ratio is near 0. We note that the linear volume
depolarization ratio is predominantly influenced by the sphericity of the dust particles
(e.g., Ansmann et al., 2003). Therefore, the polarization is sensitive to aspherical
particles, such as ice and dust. In a large number of studies, depolarization acts as a
criteria to distinguish clouds, aerosols, cloud phases, and aerosol types, especially for
dust.
The color ratio is defined as the ratio of the backscatter coefficient at 1064 nm to
that of 532 nm. The expression for this is as follows:

$$x(r) = \beta_{1064}(r)/\beta_{532}(r) \quad (2)$$

The color ratio is an indicator of the particle size. A large value represents a large
particle, and a small value represents a small particle. The color ratio is an indicator of
the particle's variable scattering of light across the available spectra and can be used
to distinguish clouds, aerosols and type of clouds. Meanwhile, the color ratio
represents the particle size. When the color ratio is large, the radius of the particle is
large, otherwise the radius is small. The color ratio is sensitive to the particle





orientation, particle shape and particle size. Because the Lidar coefficients at 532 nm
and 1064 nm are different, the color ratios derived from these coefficients show some
difference from those of other studies.

When considering the Lidar signal, the general rules used in these classifications

are as follows: if the linear volume depolarization ratio is high, then the layer is dust
dominated; if the linear volume depolarization ratio is low and the color ratio is high,
then the layer is pollution dominated; and if the linear volume depolarization ratio is
somewhere in the middle and the color ratio is high, the layer should be a mixture of
dust and pollution (and possibly other types of aerosols) (Liu et al., 2008b).

Then, according to the surface weather record and boundary layer height, a

subtype of dust aerosols (pure dust or transported anthropogenic dust) can then be
identified. According to the maximum standard deviation technique first developed by
Jordan et al. (2010), the PBL is derived using the NIES 532-nm attenuated backscatter.
Liu et al. (2015) proved that the results of the PBL height values derived from the
NIES Lidar were coincident with the ECMWF observations. Because the dust events
always occurred within the PBL and the long-range transportation related to the
westerly wind occurred above the PBL, the transported anthropogenic dust is above
the PBL, and the pure dust is within the PBL. The main cases of the pure dust and
transported anthropogenic dust are listed as follows. Case I: if there exists floating
dust, blowing dust or dust storms in the records of the surface weather stations and the
dust layer is within the PBL, the dust is regarded as pure dust. Case II: if there is no
relation of the dust to the surface weather record and the dust layer is above the PBL,





the dust layer is also regarded as pure dust that has been transported during long-range
prevailing winds. Case III: if there is no relation of the dust to the surface weather
record and the dust layer is in the PBL, the dust layer is regarded as transported
anthropogenic dust that has been transported to the SACOL station and mixed with
other anthropogenic aerosols during its transport.
From October 2009 to June 2013, there are 40 days and 451 days showing pure
dust and transported anthropogenic dust, respectively, and the sample numbers are
2709 and 32203, respectively.
**4 Case studies**
**4.1 Pure dust case**
As shown in Fig. 2, Lidar signals from the L2S-SM-II dual-band polarization
Lidar of SACOL together with HYSPLIT MODEL were used to distinguish the types
of dust. Lidar signals dependent on height and time were used to distinguish dust from
clouds and air molecules. The values of the attenuated backscatter coefficient, linear
volume depolarization ratio and color ratio of the dust are smaller than those of clouds
and greater than those of air molecules. Therefore, dust is separated from clouds and
air molecules. Then, the back trajectories from the HYSPLIT MODEL were used to
show the origins of the dust. By introducing the PBL derived from the backscatter
coefficient at 532 nm, we can regard dust within the PBL from the source regions as
pure dust, while the dust above the PBL is from cities, croplands and other
anthropogenic land surfaces and is transported anthropogenic dust. Lüthi et al. (2014)
believed that the attenuated backscatter coefficients at 532 nm were located within the





ranges of 0.0008-0.0016/km/sr, 00016-0.0044/km/sr and 0.0044-0.0072/km/sr,
corresponding to low, medium, and high aerosol concentrations. On the basis of
CALIPSO's algorithm, aerosols whose linear volume depolarization ratios are greater
than 0.075 were identified as dust (Liu et al., 2005).

Fig. 3 presents the dust case measured by the NIES Lidar on 19 October 2009.

The heights in Fig. 3 and Fig. 5 are the heights above ground level. Generally, the
NIES Lidar products indicate aerosols with green-yellow-orange color schemes and
clouds with white-gray color schemes. As show in Fig. 3, a layer (dust layer) is
detected at the height of 0-3 km. The total attenuated backscattering coefficient at 532
nm, and the linear volume depolarization ratio range from 0.0015-0.006/km/sr and
0.06-0.3, respectively, which indicate that dust particles are the main components of
this layer. Additionally, there is floating dust in the surface weather record. The black
dotted line indicates the PBL heights. As shown in Fig. 3, the dust layer is within PBL.
Therefore, the dust layer in this case is regarded as pure dust.

Additionally, three-day-back-trajectory simulations produced with the

HYSPLIT-4 model have been used to explore the most likely sources and
transportation routes of the dust events. The HYSPLIT-4 transport model
(fourth-generation of HYSPLIT model) provided by the NOAA Air Resources
Laboratory is used to calculate the simple air-parcel trajectories with interpolated
meteorological fields. The 6-h-interval final archive data are generated from the
NCEP (National Centers for Environmental Prediction) Global Data Assimilation
System (GDAS) reanalysis 3-dimensional meteorological fields. According to the





results, if dust aerosols from deserts are directly transported to SACOL by the
westerly winds, these dust aerosols are classified as pure dust. Otherwise, if they are
transported by easterly winds, the dust aerosols will pass through some cities and be
heavily influenced by human activities. In these circumstances, the dust aerosols from
the dust source regions would mix with urban pollution from other local areas; thus,
the mixture is classified as transported anthropogenic dust.

The result of the back-trajectory simulations of this case is shown in Fig. 4. The

dust trajectory starts at SACOL and is marked with a black star. The trajectories are
marked with different colors indicating starting points at different altitudes, and the
altitudes of the air-entrained dust particles during their transport are provided at the
bottom of Fig. 4. The dust aerosols detected at SACOL originate from the neighboring
Taklamakan Desert. During their transportation, few human activities are present in
their pathway. Combined with Fig. 3 and the surface weather record, these results
suggest that the aerosols are pure dust.
**4.2 Transported anthropogenic dust case**

Similarly, Fig. 5 presents the dust case measured by the NIES Lidar on 31 July

2010. As shown in Fig. 5, a dust layer is detected at a height of 0-2 km. The total
attenuated backscattering coefficient at 532 nm and the linear volume depolarization
ratio range from 0.0015-0.006/km/sr and 0.06-0.3, respectively, which indicate that
dust particles are the main components of this layer. Additionally, there is no related
record from the surface weather record. The black dotted line in Fig. 5 indicates the
PBL height. Thus, we can see that the dust layer is within the PBL. Therefore, the dust



layer is classified as transported anthropogenic dust.

The back-trajectory simulation is shown in Fig. 6 and suggests that the dust

aerosols detected at SACOL originated from Mongolia. During their transport, there
were many human activities that occurred along their path over Baotou and Yulin
cities. Taking into account the weather conditions and observation times combined
with Fig. 3 confirms that these aerosols are transported anthropogenic dust that were
mixed with anthropogenic emissions from cities.
**5. Comparison of the optical properties of two types of dust**

A histogram of the linear volume depolarization ratios of pure dust and

transported anthropogenic dust is shown in Fig. 7. The statistical results of the
frequency distributions of the linear volume depolarization ratios for pure dust and
transported anthropogenic dust show that the mean depolarization ratios of pure dust
and transported anthropogenic dust are 0.249 and 0.173, respectively; the skewness
coefficients are 1.315 and 0.038 for transported anthropogenic dust and pure dust,
respectively; and the kurtosis coefficients are -0.504 and 0.971 for transported
anthropogenic dust and pure dust, respectively. Additionally, the peak values are
approximately 0.275 for pure dust and approximately 0.095 for transported
anthropogenic dust. Freudenthaler et al. (2009) and Wandinger et al. (2010) both
found that the particle linear depolarization ratio was approximately 0.3 during
SAMUM–1 and SAMUM–2, respectively, which is consistent with our results. From
the results above, we can see that the depolarization of pure dust is greater than that of
transported anthropogenic dust, which means that pure dust is more spherical. The



reason why the depolarization of pure dust is greater than that of transported
anthropogenic dust is that during its transportation, dust is mixed with smoke or
anthropogenic aerosols, which makes the mixed aerosol nearly spherical. Specifically,
the results show that during its transportation, dust can be fully mixed with inorganic
salt (Sun et al., 2005; Shen et al., 2007; Fan et al., 1996), pollution elements such as
Se, Ni, Pb, Br, Cu (Zhang et al., 2005), black carbon (Kim et al., 2004), VOCs and
polyaromatic hydrocarbon (Hou et al., 2006), thus becoming anthropogenic dust.

If there is a threshold to distinguish pure dust and transported anthropogenic dust,

the total frequency whose linear volume depolarization ratio is larger than the
threshold is considered to be a misclassification for transported anthropogenic dust,
and those smaller than the threshold are considered to be a misclassification for pure
dust. In this way, a 0.2 linear volume depolarization ratio could be used as a threshold
for distinguishing pure dust and transported anthropogenic dust in other detections.
Using this simple classification, the misclassifications of pure dust and transported
anthropogenic dust are 27.6% and 28.0%, respectively. Meanwhile, the total
misclassification remains at a low level. Although most of the pure dust and
transported anthropogenic dust can be classified using the linear volume
depolarization ratio threshold, the overlapping value between 0.16 and 0.23 may
indicate ambiguous values for distinguishing pure dust and transported anthropogenic
dust via the linear volume depolarization ratio approach alone. Some effort is needed
to reduce misclassification.

Happily, this threshold is consistent with that of CALIPSO (Liu et al., 2005), but



is slightly smaller than the 0.23 from the results of Huang et al. (2015). In his research,
different dust aerosols were distinguished based on their geographic locations, namely,
dust aerosols (including pure dust and transported anthropogenic dust) from northern
China are classified as transported anthropogenic dust, and dust aerosols from the
Taklamakan Dessert are classified as natural dust. In this case, anthropogenic dust is a
part of natural dust and is influenced by human activity. During its long-range
transport, anthropogenic dust would mix with other aerosols and absorb water vapor
in the air. Therefore, the transported anthropogenic dust is more spherical than the
anthropogenic dust in northern China. Additionally, our results concerning the linear
volume depolarization ratio are smaller than those of Huang et al. (2015).

A histogram distribution of the color ratios for pure dust and transported

anthropogenic dust is shown in Fig. 8. The statistical results indicate that the mean
color ratios for pure dust and transported anthropogenic dust are 0.8 and 1.2,
respectively. The skewness coefficients are 2.9 and 2.1 for transported anthropogenic
dust and pure dust, respectively, and the kurtosis coefficients are 10.6 and 6.5 for
transported anthropogenic dust and pure dust, respectively. There are two peaks for
pure dust, the larger of which is 0.8, which represents the large dust particles in the
local areas during dusty days. The smaller one is 0.25, which represents the smaller
dust particles transported from the remote dust sources. The peak value for the
transported anthropogenic dust is approximately 0.5. From these results, we can see
that the color ratio of the transported anthropogenic dust is generally greater than that
of the pure dust, which means that the transported anthropogenic dust is larger. The





reason why the color ratio of transported anthropogenic dust is greater than that of
pure dust is that the dust is mixed with smoke or anthropogenic aerosols during its
transport, causing slight growth of the mixed aerosol. In the source regions, the color
ratios of dust particles are between 0.7-1.0 (Huang et al., 2007; He et al., 2015).
Huang et al. (2007) found the mean color ratio of the frequently observed dust
aerosols at heights of 4-7 km over the Tibet Plateau in the summer to be 0.83.

Zhou et al. (2013) found the relationship between the layer-integrated attenuated

backscattering coefficient and the layer-integrated depolarization ratio to distinguish
dusts, water clouds and ice clouds. Single scatterings by water droplets do not
depolarize backscattered light, but multiple scattering events do tend to depolarize
Lidar signals within water cloud. Thus, the layer-integrated depolarization ratios of
water clouds show considerably large values and increase with the layer-integrated
attenuated backscattering coefficient. The ice cloud that contains a large number of
randomly oriented ice particles corresponds to small attenuated backscattering
coefficients and high depolarization ratio values, while those containing horizontally
oriented ice crystals that could lead the presence of specular reflections show high
attenuated backscattering coefficients and small depolarization ratios. Dust is more
widely distributed with low backscattered light values and a wide range of
depolarization ratios. The obviously different distributions of dusts, water clouds, and
ice clouds can be used to identify these features. Here, we attempt to find the
attenuated backscattering coefficient and linear volume depolarization ratio
relationship between pure dust and transported anthropogenic dust. Fig. 9 and Fig. 10



depict the relationship between the attenuated backscattering coefficient and linear
volume depolarization ratio as well as the attenuated backscattering coefficients and
color ratios for pure dust and transported anthropogenic dust.
Fig. 9 shows the percentage of occurrences of pure dust and transported
anthropogenic dust in a 0.02*0.0008/km/Sr pixel. As shown in Fig. 9, the range of
attenuated backscattering coefficients is 0.0009 – 0.0073 /km/Sr and the range of
linear volume depolarization ratios is 0.06 – 0.42 for both pure dust and transported
anthropogenic dust. The distribution of pure dust seems to be symmetric, and the axis
of symmetry is at about x=0.26. The pure dust is concentrated in the middle-right
section, indicating that the attenuated backscattering coefficient and linear volume
depolarization ratio are relatively large. In contrast, the distribution of transported
anthropogenic dusts also seem to be symmetric, and the axis of symmetry is a straight
line whose slope is approximately 0.015. The transported anthropogenic dust is
concentrated in the lower-left corner, which means that the attenuated backscattering
coefficient and linear volume depolarization ratio are relatively small. Compared with
the distribution of peaks for pure dust, that for transported anthropogenic dust is
obviously shifted to the left. Among these peaks for pure dust, the minimum and
maximum values of the linear volume depolarization ratios are 0.16 and 0.34,
respectively, while for transported anthropogenic dust, the minimum and maximum
values of the linear volume depolarization ratios are 0.08 and 0.18, respectively. The
linear volume depolarization ratio of pure dust is greater than that of transported
anthropogenic dust, and the overlapping section is very small.



Next, we attempt to find the attenuated backscattering coefficient and color ratio
relationship for pure dust and transported anthropogenic dust and then use it to detect
different dust aerosols from satellite observations. Fig. 10 shows the percentage of
occurrences of pure dust and transported anthropogenic dust in a 0.1*0.0008 pixel. As
shown in Fig. 10, the range of attenuated backscattering coefficients is 0.0009 –
0.0057 /km/Sr and the range of color ratios is 0.1 – 1.5 for both pure dust and
transported anthropogenic dust. However, the obvious difference is that the range of
the color ratios for pure dust is not wider than that of transported anthropogenic dust.
The distribution of pure dust seems to be symmetric, and the axis of symmetry is a
straight line. The pure dust is concentrated in two sections (the upper-left portion and
lower-right portion), indicating that when the color ratio is small, the attenuated
backscattering coefficient is large, and when the color ratio is large, the attenuated
backscattering coefficient is small. The two sections observed for pure dust
correspond to small dust particles transported from remote source regions and large
particles transported from local areas. In contrast, the distribution of the transported
anthropogenic dust also seems to be symmetric, and the axis of symmetry is a straight
line whose slope is less than that for pure dust. The transported anthropogenic dust
distribution is concentrated in the lower-middle zone, indicating that the attenuated
backscattering coefficient is relatively small, and the color ratio is near the middle of
the possible values. Compared with the distribution of extremes for pure dust, that for
transported anthropogenic dust is distinctly set in the middle. Among those extrema
for pure dust located in the upper-left portion of the distribution, the minimum and



maximum values of the color ratios are 0.2 and 0.4, respectively, and those for pure
dust located in the lower-right portion of the distribution show minimum and
maximum values of the color ratios are 0.7 and 0.9, respectively. Meanwhile, for the
transported anthropogenic dust, the minimum and maximum values of the color ratios
are 0.4 and 0.6, respectively. On average, the color ratios of the transported
anthropogenic dust are greater than those of pure dust.
**6 Conclusions and discussion**
As we discussed above, pure dust and transported anthropogenic dust can be
distinguished by using a combination of ground-based L2S-SM-II dual-band
polarization Lidar data, surface weather station records and PBL heights. Contrasting
the frequency distributions of the linear volume depolarization ratios of two different
kinds of dust, we find the following: the mean linear volume depolarization ratios of
pure dust and transported anthropogenic dust are 0.249 and 0.173, respectively; the
maximum linear volume depolarization ratios of pure dust and transported
anthropogenic dust are 0.275 and 0.095, respectively. The mean value of pure dust is
greater than that of anthropogenic dust, which means that the pure dust is more
spherical, and based on the relationship of misclassification of pure dust and
transported anthropogenic dust verses depolarization, a threshold of 0.2 is chosen to
classify the two different kinds of dust. By contrasting the frequency distribution of
the color ratios of two different kinds of dust, we find the following: the mean color
ratios of pure dust and transported anthropogenic dust are 0.8 and 1.2, respectively;
the maximum value of the color ratio of transported anthropogenic dust is 0.5, but



there are two maxima for pure dust: the smaller is 0.25, and the larger is 0.8. The
mean value of the transported anthropogenic dust is greater than that of pure dust,
which means that transported anthropogenic dust is larger. The results of the
relationship between the attenuated backscattering coefficient and the linear volume
depolarization ratio of pure dust and transported anthropogenic dust show that the
transported anthropogenic dust is concentrated in the lower-left corner of the overall
distribution, which means the linear volume depolarization ratio is relatively small; in
contrast, the pure dust is concentrated in the right section of its distribution, implying
that the linear volume depolarization ratio is relatively large. The results of the
relationship between the attenuated backscattering coefficient and the color ratio of
pure and transported anthropogenic dusts show that there are two maxima for pure
dust: one is shown in the upper-left portion of Fig. 10 and corresponds with a small
color ratio and a large attenuated backscattering coefficient, while the other is shown
in the lower-right portion of Fig. 10 and corresponds with a large color ratio and small
attenuated backscattering coefficient. The two peaks of pure dust represent the small
dust particles transported from the remote source regions by the prevailing wind and
the large particles transported from local areas during dusty days. However, the color
ratio and attenuated backscattering coefficient for the transported anthropogenic dust
are uniformly distributed.
The dust particles transported by the prevailing winds are relatively small and
spherical, while the dust particles transported during dusty days are relatively large
and aspherical. If there are no dust events in the local regions, the dust particles are



usually transported anthropogenic dust. Therefore, the transported anthropogenic
dusts are relatively large and, owing to mixing with other types of aerosols or
anthropogenic pollution, these dust particles have relatively regular shapes (Huang et
al., 2007).

Xie et al. (2008) continuously measured aerosol optical properties with the NIES

compact Raman Lidar over Beijing, China, from 15 to 31 December 2007. Their
results indicated that the total linear volume depolarization ratio was mostly below 10%
during a pollution episode, whereas it was greater than 20% during the Asian dust
episode. The average total linear volume depolarization ratio of the nonspherical
mineral dust particles was 19.54±0.53%.

Huang et al. (2010) conducted an intensive spring aerosol sampling campaign

over northwestern and northern China as well as over a megacity in eastern China
during the spring of 2007 to investigate the mixing of Asian dusts with pollution
aerosols during their long-range transports. The western dusts were less polluted than
the other two dust sources. The western dusts contained relatively small amounts of
anthropogenic aerosols and were mainly derived from the Taklimakan Desert, which
is a paleomarine source. The northwestern dust had considerable chemical reactivities
and mixings with the sulfur precursors emitted from the coal mines along the path of
their long-range transport. The northeastern dust that reached Shanghai had high
acidity and became a mixed aerosol via its interactions with other dust, local
pollutants, and sea salts.

Asian dust is often mixed with air pollution aerosols during its transport.




Sugimoto et al. (2015b) studied the internally mixed Asian dust with air pollution
aerosols using a polarization optical particle counter and a polarization-sensitive
two-wavelength Lidar. The results showed that the backscattering linear volume
depolarization ratio was smaller for all particle sizes in polluted dust. The
backscattering color ratio of the polluted dust was comparable to that of pure dust, but
the linear volume depolarization ratio was lower for polluted dust. In addition, coarse
nonspherical particles (Asian dust) almost always existed in the background, and the
linear volume depolarization ratio showed seasonal variations with a lower linear
volume depolarization ratio in the summer. These results suggest that background
Asian dust particles are internally mixed during the summer.

With the help of surface weather station data, observations and PBL heights,

Lidar data can be used to identify pure dust and transported anthropogenic dust via
their optical properties. Then, the optical properties of pure dust and transported
anthropogenic dust can be analyzed. Last, by combining the linear volume
depolarization ratio–attenuated backscattering coefficient relationship, the color ratio–
attenuated backscattering coefficient relationship, the threshold for the linear volume
depolarization ratio and the peak values for the color ratios, our ability to identify
different dust aerosols will be greatly improved. Studies of the optical properties of
pure dust and transported anthropogenic dust using ground-based Lidar would be
highly beneficial for detecting dust using satellite data and would improve our ability
to model dust. Thus, these studies can improve our understanding of the impacts of
Asian dust on regional and global climate change as well as providing information to



help estimate the influence of human activities on the climate system.



***Acknowledgements.*** Supported by the National Fund Committee Innovation Group
(Grant No. 41521004), General Program (Grant No. 41775021), National Natural
Science Foundation of China (Grant No. 41305026)  and the National Natural Science
Foundation of China (Grant No. 41375032). Ground-based Lidar data was obtained
from the Semi-Arid Climate and Environment Observatory of Lanzhou University
(SACOL). The surface station weather data were obtained from the China
Meteorological Data Sharing Service System.



























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





**Figure captions:**

Figure 1. Spatial distribution of dust event in China, color represent the number of dust event, the locations of SACOL is shown in green pentagram, the nearby dust source (Taklimakan, Gobi) is also shown.

Figure 2. Structure of L2S-SM-II dual band depolarization lidar at SACOL (Zhou, et al, 2013).

Figure 3. Distribution of attenuated backscattering coefficient (a), linear volume depolarization ratio (b) and color ratio (c) measured by SACOL NIES on 31 March 2010.

Figure 4. Three-day back trajectories of air parcels passing by SACOL on 31 March 2010 by using NOAA HYSPLIT Model.

Figure 5. Distribution of attenuated backscattering coefficient (a), linear volume depolarization ratio (b) and color ratio (c) on 31 July 2010 by using SACOL NIES.

Figure 6. Six-day back trajectories of air parcels passing by the SACOL on 31 July 2010 by using NOAA HYSPLIT Model.

Figure 7. Comparison of the frequency distribution of linear volume depolarization ratio for pure dust (blue) and transported anthropogenic dust (red).

Figure 8. Comparison of the frequency distribution of color ratio for pure dust (blue) and transported anthropogenic dust (red).

Figure 9. Relationship between backscatter coefficient and linear volume depolarization ratio for (a) pure dust and (b) transported anthropogenic dust. The colors represent the percentage in each 0.02*0.0008 box and the value is scaled by 100.

Figure 10. Relationship between backscatter coefficient and color ratio for (a) pure dust and (b) transported anthropogenic dust. The colors represent the percentage in each 0.02*0.0008 box and the value is scaled by 100.



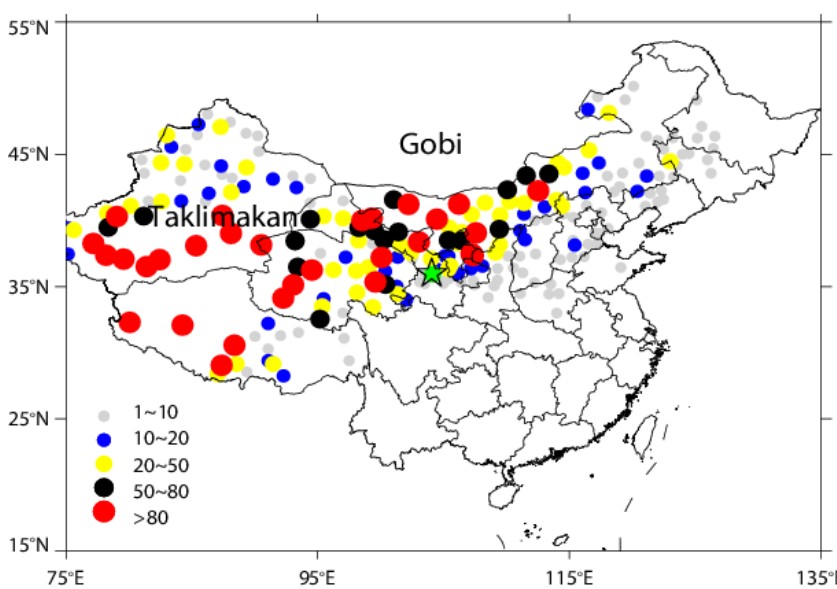


Figure 1. Spatial distribution of dust event in China, color represent the number of
dust event, the locations of SACOL is shown in green pentagram, the nearby dust
source (Taklimakan, Gobi) is also shown.



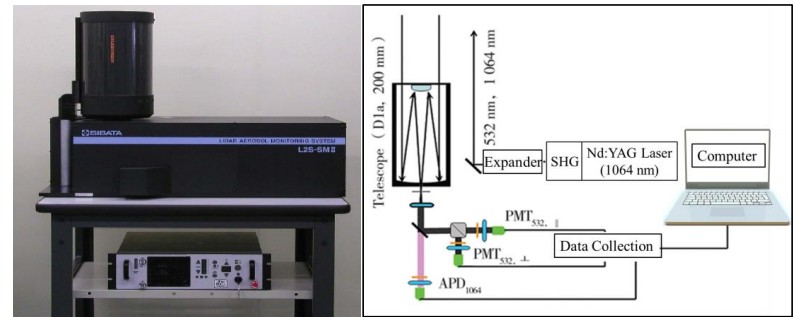

Figure 2. Structure of L2S-SM-II dual band depolarization lidar at SACOL (Zhou, et al, 2013).





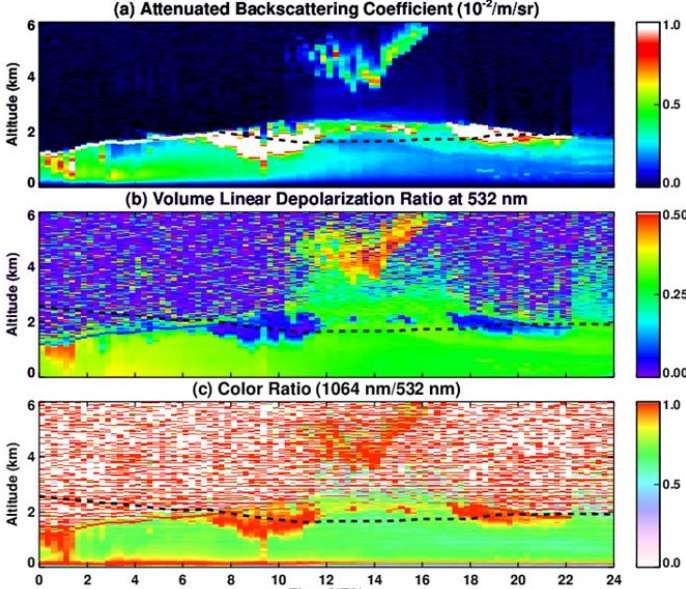

Figure 3. Distribution of attenuated backscattering coefficient at 532nm (a), linear
volume depolarization ratio (b) and color ratio (c) measured by SACOL NIES on 31
March 2010. The black dotted line indicates NIES lidar PBL height via maximum
standard deviation method (same as in the Huang et al., 2015).







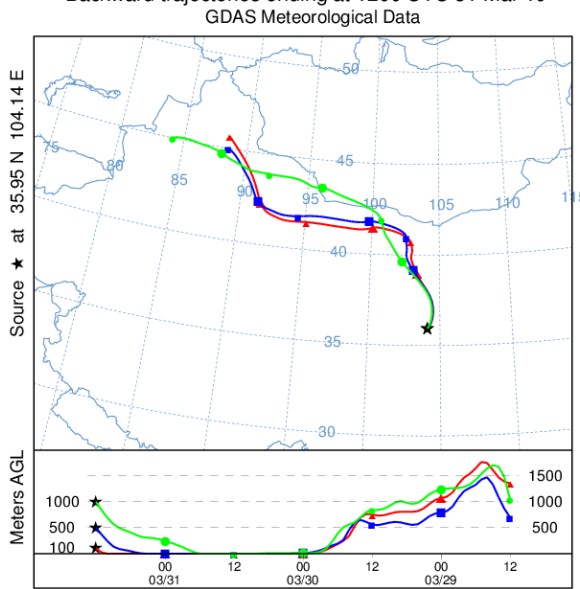

Figure 4. Three-day back trajectories of air parcels passing through SACOL on 31 March 2010 by using NOAA HYSPLIT Model.





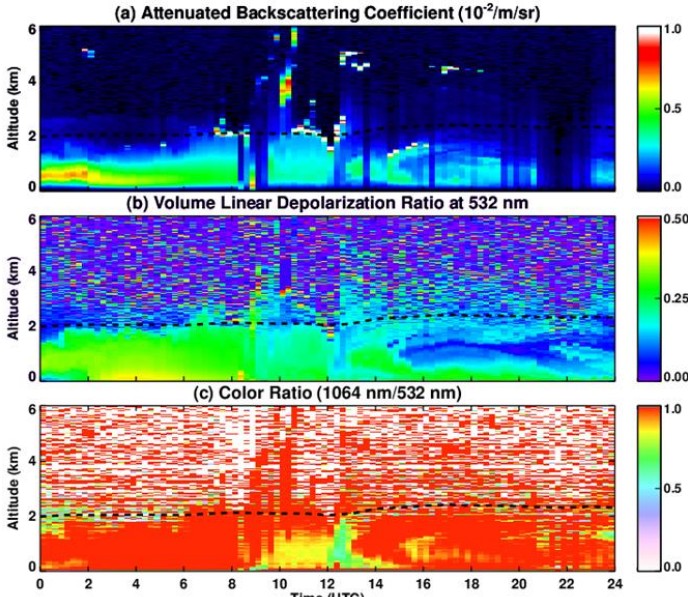

Figure 5. Distribution of attenuated backscattering coefficient at 532nm (a), linear
volume depolarization ratio (b) and color ratio (c) on 31 July 2010 by using SACOL
NIES. The black dotted line indicates NIES lidar PBL height via maximum standard
deviation method.






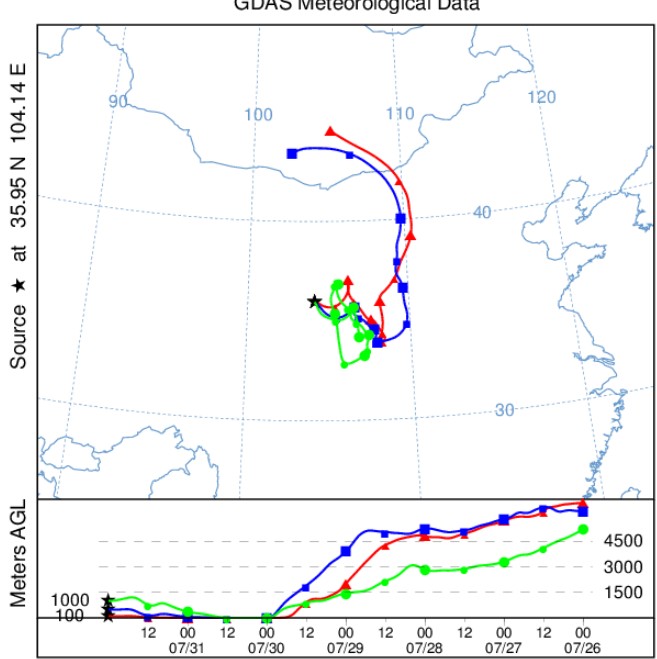


Figure 6. Six-day back trajectories of air parcels passing through the SACOL on 31
July 2010 by using NOAA HYSPLIT Model.







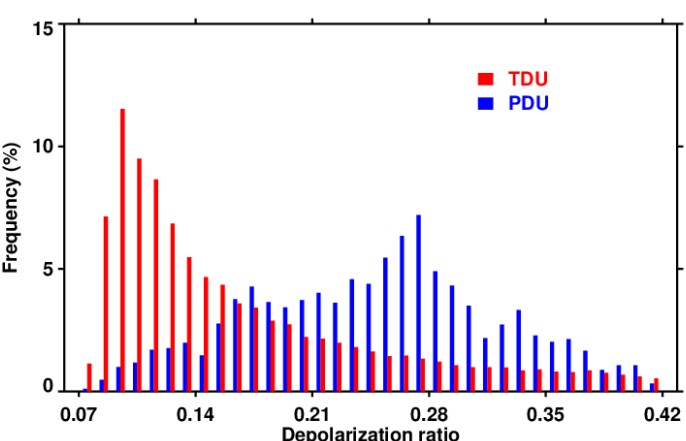

Figure 7. Comparison of the frequency distribution of linear volume depolarization
ratio for pure dust (blue) and transported anthropogenic dust (red).




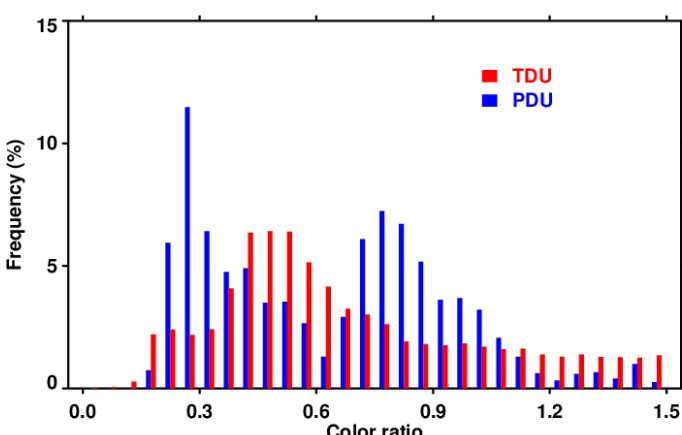


Figure 8. Comparison of the frequency distribution of color ratio for pure dust (blue)
and transported anthropogenic dust (red).














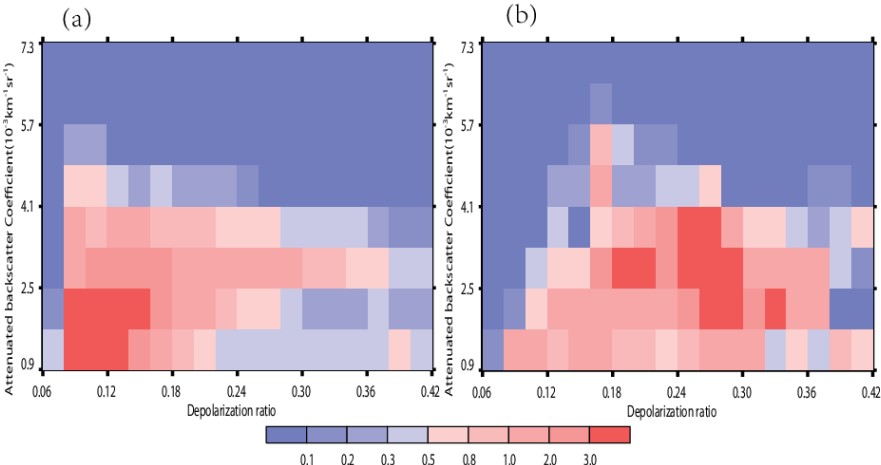


Figure 9. Relationship between backscatter coefficient and linear volume depolarization ratio for (a) pure dust and (b) transported anthropogenic dust. The colors represent the percentage in each 0.02*0.0008 box and the value is scaled by 100.






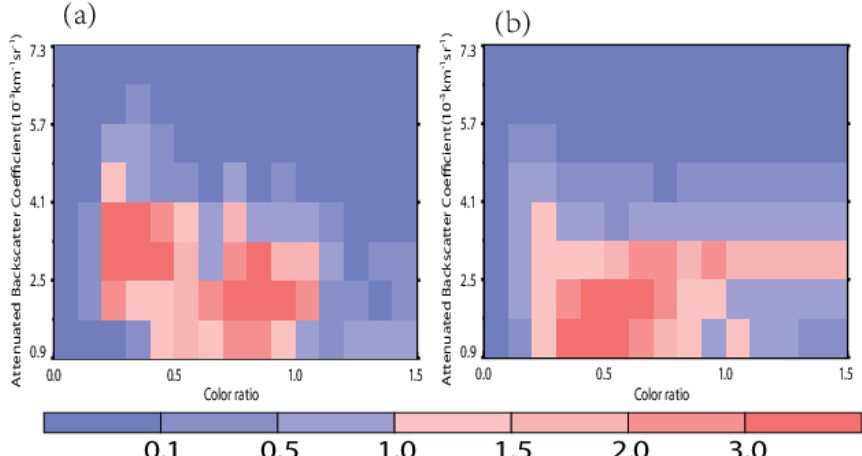


Figure 10. Relationship between backscatter coefficient and color ratio for (a) pure
dust and (b) transported anthropogenic dust. The colors represent the percentage in
each 0.02*0.0008 box and the value is scaled by 100.