# Peer review of "Comparison of the optical properties of pure and transported anthropogenic dusts measured by ground-based Lidar"

_Atmospheric Chemistry and Physics, 2017_

## Referee Comment (RC1) · Anonymous Referee #3 · 30 Dec 2017

General comments

The authors attempted to establish one or more threshold for distinguishing polluted dust from pure dust using their optical properties (i.e., total attenuated backscattering coefficient, depolarization ratio and color ratio) measured by a ground-based Lidar. They concluded that depolarization ratio threshold of 0.2 could be used to differentiate the polluted and pure dust. The authors presented a good literature review. However, the study does not give much new insight into the study topic. Generally, the manuscript was poorly written and exhibits some mistakes, which largely hampers its readability. Numerous adjoining sentences are repeated (see the specific comments). Moreover, I

have some major concerns with the methods and explanations in the manuscript.

1. The 'anthropogenic dust' is defined in a confusing way. The definition of the anthropogenic dust in the literatures (Tegen and Fung, 1995; Huang et al., 2015) is clear, that is the dust produced by human activities on disturbed soils. I cannot understand why the authors proposed a new definition for it (Lines 94-95). It seems that the anthropogenic dust in the manuscript means the dust (taking no account of its source) mixed with pollutants. Why not simplify it to "polluted dust"?

2. Can the subtype of dust aerosols be identified using the surface weather record and boundary layer height? I doubt. Firstly, the PBL height derived from the Lidar may be in low accuracy during the dusty days owing to the impact of the dust layer. Secondly, the PBL height shown in Figs.3 and 5 doesn't have a diurnal variation, why? Thirdly, according to figure.6, the authors claimed that the dust aerosols detected at SACOL originate from the Mongolia (Lines 292-293). Noting that the air parcels passed through Mongolia at an altitude of much more than 4500m, thus the dust is unlikely to originate from the Mongolia. Fourthly, still according to figure.6, the authors concluded that the dust is mixed with anthropogenic pollution when passing through Baotou and Yulin city. Again, the air parcels passed through Baotou and Yulin at an altitude of more than 4500m, the dust is unlikely to be mixed with anthropogenic pollution. Moreover, the dust may enter the atmosphere when the air parcels were in contact with the surface (starting at 00:00 UTC, 31 Mar 2010 in Fig.4, and 00:00 UTC, 30 Jul 2010 in Fig.6) or even prior to the start date of the back-trajectory simulations, doesn't it? The results will be more reliable if both the pathway and the altitude of the air parcels were considered.

3. When was the first dust case detected? On 19 October 2009 (see line 253) or 31 March 2010 (see line 743)?

4. According to the manuscript, there were two types of pure dust: a) dust layer within the PBL and recorded by the weather stations; b) dust layer above the PBL and not recorded by the weather stations. It seems that the later one is more likely to be "pure

dust", is there any different between their optical properties?

5. With regard to Figs. 1 and 2, the discussions in the main text did not match the plots. Actually, the Fig.2 rather than Fig.1 shows the structure of the Lidar. The Fig. 1 was not discussed in the main text.

6. Color ratio is an indicator for particle size. A large value represents big particle and a small value represents small particle. Generally speaking, anthropogenic aerosols are mainly composed of fine mode particles, why it has a large color ratio (see line 219)?

Specific comments

1. Check the order of the subtitle of Fig 9. The right panel should be the results of pure dust.

2. Mistake in lines 309-310: 'From the results above we can see the depolarization of pure dust is larger than that of anthropogenic dust which means the pure dust is more sphere.' 'the pure dust is more sphere' should be 'the anthropogenic dust is more sphere'

3. Mistake in lines 426-427: 'The mean value of pure dust is larger than that of anthropogenic dust, which means that the pure dust is more a spherical' 'the pure dust is more a spherical' should be 'the anthropogenic dust'.

4. Lines 103-104: remove 'mixed with the anthropogenic dust'.

5. Lines 143-147: there is no essential difference between 'the environment of the mountain top is almost natural, and is rarely affected by human activity' and 'building at the top of the mountain, the influence of houses and human activity is escaped', delete the later one.

6. Line 359, change 'found' to 'used'.

7. Line 474-477, duplicate sentence. Remove 'Results showed that the backscattering depolarization ratio was smaller for all particle sizes in polluted dust.'

Please also note the supplement to this comment:
https://www.atmos-chem-phys-discuss.net/acp-2017-1000/acp-2017-1000-RC1-supplement.pdf
* * *

---

## Referee Comment (RC2) · Anonymous Referee #1 · 11 Jan 2018

General Comments:

"Comparison of optical properties of pure dust and transported anthropogenic dusts measured by ground-based Lidar" describes two cases and statistical analysis of pure and anthropogenic dust based on the polarization sensitive lidar observations. The depolarization ratio by lidar is an important parameter for dust studies, but authors utilize "volume depolarization ratio" which represents non-sphericity of particles in qualitative manner because it depends on scattering ratio. At least "particle depolarization ratio" should be used to describe the characteristics of dust quantitatively. Also authors should clearly distinguish the mixing of dust and pollutant "internally" or "externally"

throughout the study. From these points of view, this manuscript must be fundamentally revised before publication.

Specific Comments:

Spelling "Lidar" is not common. Just "lidar" is adequate.

L128, what is the time resolution of surface weather data?

L134, refer Figure 1.

L190, Winker et al.(2009, not 2006) compared the detectors in CALIOP, not the lidar in SOCAL.

L201, the depolarization ratio represents statistical properties of particles in the observed volume, not a single particle.

Eq(2), how did author retrieve beta1064? By Fernald method?

L228, does low DEP and high CR correspond to pollution? It seems coarse sphere, like sea salt.

L240, if dust is reported at stations and dust layer is detected above PBL by lidar, is it pure or transported?

L318, what is the target of statistical analysis? All data during October 2009 and June 2013? Or, some restriction by scattering ratio? What is the height range?

L367, what is the physical meaning of skewness and kurtosis for histograms?

Figure 1, describe the time period in which the number of dust events were counted.

Figure 3 and 5, unit for panel (a) is unnatural. Is it 10ˆ-2/km/sr?

Figure 3, PBL height at 0 UTC was above the cloud layer. How lidar can detect it without effective signal?

Figure 4 and 6, all trajectories touch the ground. Are these paths reliable?

Figure 9, (b) for pure and (a) for anthropogenic dust.

Technical Corrections:

L53, L58, L89 etc, unify the usage of "," and ";".

References, J. Quant. Spectrosc. Radiat. Transfer

---

## Referee Comment (RC3) · Anonymous Referee #2 · 2 Feb 2018

Unfortunately, the paper is unacceptable. The location of the lidar observations (SACOL site) is excellent. The lidar data set is probably of high quality. So I would like to encourage the authors to resubmit the paper after considering my suggestions.

The main reason for rejection is that the authors fail to provide a clear definition and thus separation of pure dust and anthropogenic dust cases. A clear definition can be done by means of the particle linear depolarization ratio. But the authors only present volume depolarization ratios. These values vary with the relative amount of dust, and thus can be low even in the case of pure dust, and large, even in the case of polluted dust. So the only way is: Compute the particle depolarization ratio and use

this parameter to distinguish polluted (or anthropogenic) and pure dust cases. If the particle depolarization ratio is > 25% one may call the event a pure dust case and if we have <25% then we may call it a polluted dust case.

Furthermore, most of the results are simply given in terms of attenuated backscatter. This quantity varies with the amount of aerosol, so with the amount of dust and/or pollution. We need the particle backscatter coefficient to describe aerosol properties with height.

The overall impression is: The paper is to 80% just based on 'opinions', and not on 'objective' facts. The lidar community dealing with dust research would be upset if this low-quality paper gets published in its present form.

The authors may want to resubmit their paper. Then the analysis must be fully based on (a) particle backscatter coefficients for 532 and 1064 nm, (and not on 532 nm attenuated backscatter) and (b) on particle depolarization ratios (and not on volume depolarization ratios). The particle depolarization ratio can be easily computed from the volume depolarization ratio and the 532 nm particle backscatter coefficient (see the cited publication of Freudenthaler 2009, or some papers from the NIES group). And then introduce a clear criterion for anthropogenic dust, based on the particle linear depolarization ratio.

---

## Author Comment (AC1) · 9 Feb 2018

1. The 'anthropogenic dust' is defined in a confusing way. The definition of the anthropogenic dust in the literatures (Tegen and Fung, 1995; Huang et al., 2015) is clear, that is the dust produced by human activities on disturbed soils. I cannot understand why the authors proposed a new definition for it (Lines 94-95). It seems that the anthropogenic dust in the manuscript means the dust (taking no account of its source) mixed with pollutants. Why not simplify it to "polluted dust"?

Sorry for my confusing definition of 'anthropogenic dust' in this manuscript. Actually in this manuscript one of the dusts I focused is transported anthropogenic dust. It is a kind

of dust that originates from dust source regions and mixes with anthropogenic polluted aerosols during transportation. We used this word 'anthropogenic dust' is mainly to strengthen the human influence and then to study the different optical properties of dust from natural source and influenced by human activities. So owing to the word 'anthropogenic dust' is not properly used, now we change it to 'polluted dust' after serious consideration.

2. Can the subtype of dust aerosols be identified using the surface weather record and boundary layer height? I doubt. Firstly, the PBL height derived from the Lidar may be in low accuracy during the dusty days owing to the impact of the dust layer. Secondly, the PBL height shown in Figs.3 and 5 doesn't have a diurnal variation, why? Thirdly, according to figure.6, the authors claimed that the dust aerosols detected at SACOL originate from the Mongolia (Lines 292-293). Noting that the air parcels passed through Mongolia at an altitude of much more than 4500m, thus the dust is unlikely to originate from the Mongolia. Fourthly, still according to figure.6, the authors concluded that the dust is mixed with anthropogenic pollution when passing through Baotou and Yulin city. Again, the air parcels passed through Baotou and Yulin at an altitude of more than 4500m, the dust is unlikely to be mixed with anthropogenic pollution. Moreover, the dust may enter the atmosphere when the air parcels were in contact with the surface (starting at 00:00 UTC, 31 Mar 2010 in Fig.4, and 00:00 UTC, 30 Jul 2010 in Fig.6) or even prior to the start date of the back-trajectory simulations, doesn't it? The results will be more reliable if both the pathway and the altitude of the air parcels were considered. When was the first dust case detected? On 19 October 2009 (see line 253) or 31 March 2010 (see line 743)?

The detection method used in this manuscript refers to the literature of Huang et al. (2015). Pure dust and transported anthropogenic dust can be distinguished by using a combination of ground-based L2S-SM-II dual-band polarization lidar data, surface weather station records and PBL heights. Firstly, we agree with you that the PBL height derived from lidar is in low accuracy during the dusty days. The height of dust

when in the dust storm is usually below the PBL and the occasion is rare that above the PBL it is small transported dust particles and below PBL it is dusty days. If there is dust storm, we didn't calculate the PBL and we regarded all the detected dust as pure dust. Secondly, the BLH, derived from soundings conducted three or four times daily in summer, tends to peak in the early afternoon, and the diurnal amplitude of PBL is higher in the northern and western subregions of China than other subregions. During a diurnal cycle, the BLH is typically shallow (a few hundred meters) at night due to the strong near-surface stability, and the PBL can be well developed and reach several kilometers in the afternoon (Guo J. et al., 2016). Guo S. et al. (2014) found a lack of diurnal variation, but a cycle of 4–7 days in the aerosol properties, indicating a reduced PBL diurnal trend during polluted periods. But the PBL height shown in Figs.3 and 5 doesn't have a diurnal variation. That's because deriving PBL using lidar is better in daytime. Daytime observations were used from CALIPSO to ensure that residual layers were not picked out in nighttime data (Liu et al., 2015). Another two cases were picked out as shown in figure 3 and 5. Third, we agree with you that the pathway and altitude should both be considered. Now we are finding appropriate cases. Last, first dust case detected is 31 March 2010. We have corrected it in the manuscript.

3. According to the manuscript, there were two types of pure dust: a) dust layer within the PBL and recorded by the weather stations; b) dust layer above the PBL and not recorded by the weather stations. It seems that the later one is more likely to be "pure dust", is there any different between their optical properties?

Yes, they are different between their optical properties. As the results of this manuscript show, the linear volume depolarization ratio of polluted that is smaller than that of pure dust which means the polluted dust is more spherical. The color ratio of polluted dust from dusty days is smaller than that of pure dust which means the polluted dust is smaller in size.

4. With regard to Figs. 1 and 2, the discussions in the main text did not match the plots. Actually, the Fig.2 rather than Fig.1 shows the structure of the Lidar. The Fig. 1

was not discussed in the main text.

Thank you for your kind reminding. As you said, in this manuscript Fig.2 shows the structure of the Lidar. The Fig.1 was not discussed in the main text and now we add this part in the manuscript.

5. Color ratio is an indicator for particle size. A large value represents big particle and a small value represents small particle. Generally speaking, anthropogenic aerosols are mainly composed of fine mode particles, why it has a large color ratio (see line 219)?

Sorry for the confusion that caused by my poor expression. The transported anthropogenic dust in this manuscript is actually polluted dust. It is a mixture of pure dust from remote dust source regions and anthropogenic polluted aerosols. So it is larger than the dust from remote dust source regions that did not mix with polluted aerosols. There are two kinds of pure dust in this manuscript, one is form remote dust source regions and one is from dusty days. Compared with the pure dust from remote dust source regions, the color ratio of polluted dust (anthropogenic aerosols, as you mentioned in the above question) is larger.

Specific comments

1. Check the order of the subtitle of Fig 9. The right panel should be the results of pure dust.

Thank you for your kind reminding. We have corrected in the corresponding position.

2. Mistake in lines 309-310: 'From the results above we can see the depolarization of pure dust is larger than that of anthropogenic dust which means the pure dust is more sphere.' 'the pure dust is more sphere' should be 'the anthropogenic dust is more sphere'

Sorry for the mistake that we have made. We have corrected it in the manuscript.

3. Mistake in lines 426-427: 'The mean value of pure dust is larger than that of anthropogenic dust, which means that the pure dust is more a spherical' 'the pure dust is more a spherical' should be 'the anthropogenic dust'.

Sorry for the mistake that we have made. We have corrected it in the manuscript.

4. Lines 103-104: remove 'mixed with the anthropogenic dust'.

Sorry for the mistake that we have made. We have corrected it in the manuscript.

5. Lines 143-147: there is no essential difference between 'the environment of the mountain top is almost natural, and is rarely affected by human activity' and 'building at the top of the mountain, the influence of houses and human activity is escaped', delete the later one.

Sorry for the mistake that we have made. We have corrected it in the manuscript.

6. Line 359, change 'found' to 'used'.

Sorry for the mistake that we have made. We have corrected it in the manuscript.

7. Line 474-477, duplicate sentence. Remove 'Results showed that the backscattering depolarization ratio was smaller for all particle sizes in polluted dust.

Sorry for the mistake that we have made. We have corrected it in the manuscript.

Reference [1] Liu, J., Huang, J., Chen, B., Zhou, T., Yan, H., Jin, H., Huang, Z. and Zhang, B.: Comparisons of PBL heights derived from CALIPSO and ECMWF reanalysis data over China, Journal of Quantitative Spectroscopy and Radiative Transfer, J Quant Spectrosc Ra, 153, 102-112, 2015. [2] Guo J, Miao Y, Zhang Y, et al. The climatology of planetary boundary layer height in China derived from radiosonde and reanalysis data[J]. Atmospheric Chemistry and Physics, 2016, 16(20): 13309. [3] Guo, S., Hu, M., Zamora, M. L., Peng, J., Shang, D., Zheng, J., Du, Z.,Wu, Z., Shao, M., and Zeng, L.: Elucidating severe urban haze formation in China, P. Natl. Acad. Sci. USA, 111, 17373–17378, 2014.

---

## Author Comment (AC2) · 9 Feb 2018

General Comments:

"Comparison of optical properties of pure dust and transported anthropogenic dusts measured by ground-based Lidar" describes two cases and statistical analysis of pure and anthropogenic dust based on the polarization sensitive lidar observations. The depolarization ratio by lidar is an important parameter for dust studies, but authors utilize "volume depolarization ratio" which represents non-sphericity of particles in qualitative manner because it depends on scattering ratio. At least "particle depolarization ratio" should be used to describe the characteristics of dust quantitatively. Also authors

should clearly distinguish the mixing of dust and pollutant "internally" or "externally" throughout the study. From these points of view, this manuscript must be fundamentally revised before publication.

Thank you for your serious review. After your suggestion, we are now calculating the particle depolarization ratio and other related work. Owing to the vertical resolution of lidar, we can observe the volume of 6 m high. So in this air column there are many dust aerosols rather than one dust aerosol, so we think the mixing state is needlessly distinguished.

Specific Comments:

1. Spelling "Lidar" is not common. Just "lidar" is adequate. Thank you for your kind reminding. We have corrected in the corresponding position.

2. L128, what is the time resolution of surface weather data? Sorry for our ignorance of that. The time resolution of surface weather data is daily.

3. L134, refer Figure 1. Thank you for your kind reminding. We have corrected in the corresponding position.

4. L190, Winker et al.(2009, not 2006) compared the detectors in CALIOP, not the lidar in SOCAL. Thank you for your kind reminding. We have corrected in the corresponding position. We referred that literature to illustrate we choose attenuated backscatter coefficient at 532 nm to discriminate clouds and aerosols, rather than 1064 nm.

5. L201, the depolarization ratio represents statistical properties of particles in the observed volume, not a single particle. Eq(2), how did author retrieve beta1064? By Fernald method? Yes, you are right. The depolarization ratio represents statistical properties of particles in the observed volume. Also, we retrieve beta1 1064 by Fernald method in Eq(2).

6. L228, does low DEP and high CR correspond to pollution? It seems coarse sphere, like sea salt. According to our results in our manuscript, the DEP of polluted dust is

relatively low compared with pure dust, but it is relatively high than that of sea salt. Because only when the DEP of aerosols is greater than 0.06, can this kind of aerosols be picked out. The CR of polluted dust is 1.1 and for sea salt it is 0.53. When we pick polluted dust, the threshold of CR is greater than 0.6.

7. L240, if dust is reported at stations and dust layer is detected above PBL by lidar, is it pure or transported? We thought it was pure dust.

8. L318, what is the target of statistical analysis? All data during October 2009 and June 2013? Or, some restriction by scattering ratio? What is the height range? We did statistical analysis to find the threshold between pure dust and polluted dust from the optical perspective and further to improve the detection of different aerosol type in numerical modeling and satellite algorithm. Not all data during October 2009 and June 2013. We have conducted strict quality control. Every case was strictly picked out using IDL source code and after that we confirmed every case personally with our eyes. The original height range is 0 to 18 km, and we choose 0-6 km above the ground.

9. L367, what is the physical meaning of skewness and kurtosis for histograms? Skewness is a measure of the direction and extent of skewness in the distribution of statistical data and is a numerical feature of the degree of asymmetry in the distribution of statistical data. The number of features characterizing the degree of asymmetry of the probability distribution density curve with respect to the mean. Intuitively, it is the relative length of the tail of the density function curve. In our results, take depolarization ratio for example, skewness of pure dust and polluted dust are greater than 0, which means they all located on the right less than the left. But the skewness of pure dust is smaller than that of polluted dust which means for pure dust the number of large values is large. Mean value alone cannot describe the distribution of pure dust and polluted dust, so we add skewness and kurtosis to help us to detect them clearly in the space-born lidar and numerical modelling.

10. Figure 1, describe the time period in which the number of dust events were

counted. The time period in Figure 1 is 2013, one year data.

11. Figure 3 and 5, unit for panel (a) is unnatural. Is it 10Ë£-2/km/sr? Thank you for your kind reminding. We have corrected in the corresponding position.

12. Figure 3, PBL height at 0 UTC was above the cloud layer. How lidar can detect it without effective signal? Thank you for your kind reminding. We have picked the cases with strict control. Now we are doing these works.

13. Figure 4 and 6, all trajectories touch the ground. Are these paths reliable? Thank you for your kind reminding. We have picked the cases with strict control. Now we are doing these works.

14. Figure 9, (b) for pure and (a) for anthropogenic dust. Thank you for your kind reminding. We have corrected in the corresponding position.

Technical Corrections:

1. L53, L58, L89 etc, unify the usage of "," and ";".

Thank you for your kind reminding. We have corrected in the corresponding position.

2. References, J. Quant. Spectrosc. Radiat. Transfer

Thank you for your kind reminding. We have corrected in the corresponding position.

---

## Author Comment (AC3) · 9 Feb 2018

General Comments:

Unfortunately, the paper is unacceptable. The location of the lidar observations (SACOL site) is excellent. The lidar data set is probably of high quality. So I would like to encourage the authors to resubmit the paper after considering my suggestions. The main reason for rejection is that the authors fail to provide a clear definition and thus separation of pure dust and anthropogenic dust cases. A clear definition can be done by means of the particle linear depolarization ratio. But the authors only present volume depolarization ratios. These values vary with the relative amount of dust, and

thus can be low even in the case of pure dust, and large, even in the case of polluted dust. So the only way is: Compute the particle depolarization ratio and use this parameter to distinguish polluted (or anthropogenic) and pure dust cases. If the particle depolarization ratio is > 25% one may call the event a pure dust case and if we have <25% then we may call it a polluted dust case. Furthermore, most of the results are simply given in terms of attenuated backscatter. This quantity varies with the amount of aerosol, so with the amount of dust and/or pollution. We need the particle backscatter coefficient to describe aerosol properties with height. The overall impression is: The paper is to 80% just based on 'opinions', and not on'objective' facts. The lidar community dealing with dust research would be upset if this low-quality paper gets published in its present form. The authors may want to resubmit their paper. Then the analysis must be fully based on (a) particle backscatter coefficients for 532 and 1064 nm, (and not on 532 nm attenuated backscatter) and (b) on particle depolarization ratios (and not on volume depolarization ratios). The particle depolarization ratio can be easily computed from the volume depolarization ratio and the 532 nm particle backscatter coefficient (see the cited publication of Freudenthaler 2009, or some papers from the NIES group). And then introduce a clear criterion for anthropogenic dust, based on the particle linear depolarization ratio.

Thank you for your serious review. We are now calculating the particle linear depolarization ratio and particle backscatter coefficients for 532 and 1064 nm. Because time is limited, we have applied for much longer time to present high quality manuscript.

---

## Author Comment (AC4) · 10 Mar 2018

General comments

The authors attempted to establish one or more threshold for distinguishing polluted dust from pure dust using their optical properties (i.e., total attenuated backscattering coefficient, depolarization ratio and color ratio) measured by a ground-based Lidar. They concluded that depolarization ratio threshold of 0.2 could be used to differentiate the polluted and pure dust. The authors presented a good literature review. However,

the study does not give much new insight into the study topic. Generally, the manuscript was poorly written and exhibits some mistakes, which largely hampers its readability. Numerous adjoining sentences are repeated (see the specific comments). Moreover, I have some major concerns with the methods and explanations in the manuscript.

Thank you for your serious review. This manuscript was intended to further prove the detection method of anthropogenic dust first proposed by Huang et al. (2015). After your suggestions, we have rewrote the manuscript.

1. The 'anthropogenic dust' is defined in a confusing way. The definition of the anthropogenic dust in the literatures (Tegen and Fung, 1995; Huang et al., 2015) is clear, that is the dust produced by human activities on disturbed soils. I cannot understand why the authors proposed a new definition for it (Lines 94-95). It seems that the anthropogenic dust in the manuscript means the dust (taking no account of its source) mixed with pollutants. Why not simplify it to "polluted dust"?

Sorry for my confusing definition of 'anthropogenic dust' in this manuscript. Actually in this manuscript one of the dusts I focused is transported anthropogenic dust. It is a kind of dust that originates from dust source regions and mixes with anthropogenic polluted aerosols during transportation. We used this word 'anthropogenic dust' is mainly to strengthen the human influence and then to study the different optical properties of dust from natural source and influenced by human activities. So owing to the word 'anthropogenic dust' is not properly used, now we change it to 'polluted dust' after serious consideration.

2. Can the subtype of dust aerosols be identified using the surface weather record and boundary layer height? I doubt. Firstly, the PBL height derived from the Lidar may be in low accuracy during the dusty days owing to the impact of the dust layer. Seco ndly, the PBL height shown in Figs.3 and 5 doesn't have a diurnal variation, why? Thirdly, according to figure.6, the authors claimed that the dust aerosols detected at SACOL originate from the Mongolia (Lines 292-293). Noting that the air parcels passed through

Mongolia at an altitude of much more than 4500m, thus the dust is unlikely to originate from the Mongolia. Fourthly, still according to figure.6, the authors concluded that the dust is mixed with anthropogenic pollution when passing through Baotou and Yulin city. Again, the air parcels passed through Baotou and Yulin at an altitude of more than 4500m, the dust is unlikely to be mixed with anthropogenic pollution. Moreover, the dust may enter the atmosphere when the air parcels were in contact with the surface (starting at 00:00 UTC, 31 Mar 2010 in Fig.4, and 00:00 UTC, 30 Jul 2010 in Fig.6) or even prior to the start date of the back-trajectory simulations, doesn't it? The results will be more reliable if both the pathway and the altitude of the air parcels were considered. When was the first dust case detected? On 19 October 2009 (see line 253) or 31 March 2010 (see line 743)?

The detection method used in this manuscript refers to the literature of Huang et al. (2015). Pure dust and polluted dust can be distinguished by using a combination of ground-based L2S-SM-II dual-band polarization lidar data, surface weather station records, PBL heights and back trajectories. Firstly, we agree with you that the PBL height derived from lidar is in low accuracy during the dusty days. If there is dust storm, we didn't calculate the PBL and regarded all the detected dust as pure dust. This will be stated clearly in the manuscript. Secondly, the PBL, derived from soundings conducted three or four times daily in summer, tends to peak in the early afternoon, and the diurnal amplitude of PBL is higher in the northern and western subregions of China than other subregions. During a diurnal cycle, the PBL is typically shallow (a few hundred meters) at night due to the strong near-surface stability, and the PBL can be well developed and reach several kilometers in the afternoon (Guo J. et al., 2016). This is the typical diurnal variation. Guo et al. (2014) found a lack of diurnal variation, but a cycle of 4-7 days in the aerosol properties, indicating a reduced PBL diurnal trend during polluted periods. In figure 3 we didn't calculate the PBL, and in figure 5, it did not show the diurnal variation. That's because deriving PBL using lidar is better in daytime and relatively clean days. Daytime observations were used from CALIPSO to ensure that residual layers were not picked out in nighttime data (Liu et al., 2015). Another

two cases were picked out as shown in figure 3 and 5. Third, we agree with you that the pathway and altitude should both be considered. Appropriate cases were shown in Figure 4 and Figure 6. Last, the time of first dust case is 19 March 2010. And we have corrected it in the manuscript.

3. According to the manuscript, there were two types of pure dust: a) dust layer within the PBL and recorded by the weather stations; b) dust layer above the PBL and not recorded by the weather stations. It seems that the later one is more likely to be "pure dust", is there any different between their optical properties?

Sorry for the unreasonable detection method that we originally used. After analyzing every lidar picture, we found that quite a few dust aerosols can reach 3 km above the ground or even higher and the optical properties of those dusts is close. So we can't use PBL to clarify them as pure dust and polluted dust. In our new detection method, we used back trajectories to help us clarify the aerosol type.

4. With regard to Figs. 1 and 2, the discussions in the main text did not match the plots. Actually, the Fig.2 rather than Fig.1 shows the structure of the Lidar. The Fig. 1 was not discussed in the main text.

Thank you for your kind reminding. As you said, in this manuscript Fig.2 shows the structure of the Lidar. The Fig.1 was not discussed in the main text and now we add this part in the manuscript.

5. Color ratio is an indicator for particle size. A large value represents big particle and a small value represents small particle. Generally speaking, anthropogenic aerosols are mainly composed of fine mode particles, why it has a large color ratio (see line 219)?

Sorry for the confusion that caused by my poor expression. The transported anthropogenic dust in this manuscript is actually polluted dust. It is a mixture of pure dust from remote dust source regions and anthropogenic polluted aerosols. So it is larger than the dust from remote dust source regions that did not mix with polluted aerosols. There

are two kinds of pure dust in this manuscript, one is form remote dust source regions and one is from dusty days. Compared with the pure dust from remote dust source regions, the color ratio of polluted dust (anthropogenic aerosols, as you mentioned in the above question) is larger.

Specific comments

1. Check the order of the subtitle of Fig 9. The right panel should be the results of pure dust.

Thank you for your kind reminding. We have corrected in the corresponding position.

2. Mistake in lines 309-310: 'From the results above we can see the depolarization of pure dust is larger than that of anthropogenic dust which means the pure dust is more sphere.' 'the pure dust is more sphere' should be 'the anthropogenic dust is more sphere'

Sorry for the mistake that we have made. We have corrected it in the manuscript.

3. Mistake in lines 426-427: 'The mean value of pure dust is larger than that of anthropogenic dust, which means that the pure dust is more a spherical' 'the pure dust is more a spherical' should be 'the anthropogenic dust'. Sorry for the mistake that we have made. We have corrected it in the manuscript.

4. Lines 103-104: remove 'mixed with the anthropogenic dust'.

Sorry for the mistake that we have made. We have corrected it in the manuscript.

5. Lines 143-147: there is no essential difference between 'the environment of the mountain top is almost natural, and is rarely affected by human activity' and 'building at the top of the mountain, the influence of houses and human activity is escaped', delete the later one.

Sorry for the mistake that we have made. We have corrected it in the manuscript.

6. Line 359, change 'found' to 'used'.

Sorry for the mistake that we have made. We have corrected it in the manuscript.

7. Line 474-477, duplicate sentence. Remove 'Results showed that the backscattering depolarization ratio was smaller for all particle sizes in polluted dust.

Sorry for the mistake that we have made. We have corrected it in the manuscript.

Reference

[1] Liu, J., Huang, J., Chen, B., Zhou, T., Yan, H., Jin, H., Huang, Z. and Zhang, B.: Comparisons of PBL heights derived from CALIPSO and ECMWF reanalysis data over China, Journal of Quantitative Spectroscopy and Radiative Transfer, J Quant Spectrosc Ra, 153, 102-112, 2015.

[2] Guo J, Miao Y, Zhang Y, et al. The climatology of planetary boundary layer height in China derived from radiosonde and reanalysis data[J]. Atmospheric Chemistry and Physics, 2016, 16(20): 13309.

[3] Guo, S., Hu, M., Zamora, M. L., Peng, J., Shang, D., Zheng, J., Du, Z.,Wu, Z., Shao, M., and Zeng, L.: Elucidating severe urban haze formation in China, P. Natl. Acad. Sci. USA, 111, 17373–17378, 2014.

---

## Author Comment (AC5) · 10 Mar 2018

General Comments:

"Comparison of optical properties of pure dust and transported anthropogenic dusts measured by ground-based Lidar" describes two cases and statistical analysis of pure and anthropogenic dust based on the polarization sensitive lidar observations. The depolarization ratio by lidar is an important parameter for dust studies, but authors utilize "volume depolarization ratio" which represents non-sphericity of particles in qualitative

manner because it depends on scattering ratio. At least "particle depolarization ratio" should be used to describe the characteristics of dust quantitatively. Also authors should clearly distinguish the mixing of dust and pollutant "internally" or "externally" throughout the study. From these points of view, this manuscript must be fundamentally revised before publication.

Thank you for your serious review. After your suggestion, we are now calculating the particle depolarization ratio and correcting other related expression. First, linear volume depolarization ratio is provided by SACOL group and is used as a part to detect pure dust and anthropogenic dust. This method is the same with Huang et al. (2015) and Liu et al. (2005). This work is mainly to further prove the detection method of anthropogenic dust in Huang et al. (2015). Second, owing to the vertical resolution of lidar, we can observe a bin of 6 meters in vertical direction. So in this air column there are many dust aerosols rather than one, so we think the mixing state is difficult to find out if we only use the NIES lidar. Compared with the CALIPSO observation, the mixing state of dust has not been considered in a good manner in the algorithm of aerosol subtype. But if we clarity the mixing state of dust aerosols, the work will be more accurate and interesting. This work is basic and it still needs to improve in the future. And we will address all those problem in the next work.

Specific Comments:

1. Spelling "Lidar" is not common. Just "lidar" is adequate.

Thank you for your kind reminding. We have corrected in the corresponding position.

2. L128, what is the time resolution of surface weather data?

Sorry for our ignorance of that. The time resolution of surface weather data is daily.

3. L134, refer Figure 1.

Thank you for your kind reminding. We have corrected in the corresponding position.

4. L190, Winker et al.(2009, not 2006) compared the detectors in CALIOP, not the lidar in SOCAL.

Thank you for your kind reminding. Winker et al. (2009) compared the detectors in CALIOP and used attenuated backscatter coefficient at 532 nm to discriminate clouds and aerosols, rather than 1064 nm. We referred that literature to illustrate that we also choose attenuated backscatter coefficient at 532 nm to discriminate clouds and aerosols, rather than 1064 nm. As your kind reminding, we have corrected in the corresponding position.

5. L201, the depolarization ratio represents statistical properties of particles in the observed volume, not a single particle. Eq(2), how did author retrieve beta1064? By Fernald method?

Yes, you are right. The depolarization ratio represents statistical properties of particles in the observed volume. We used this physical variable in our detection method, as used in Huang et al. (2015) to detect anthropogenic dust. The second question, we retrieve beta1064 in Eq(2) by Fernald method. .

6. L228, does low DEP and high CR correspond to pollution? It seems coarse sphere, like sea salt.

According to our results in our manuscript, the DEP of polluted dust is relatively low compared with pure dust, but it is relatively high than that of sea salt. Because only when the DEP of aerosols is greater than 0.06, can this kind of aerosols be picked out. So in this step sea salt is removed. On the other hand, the CR of polluted dust is 1.1 and for sea salt it is 0.53. When we pick polluted dust, the threshold of CR is greater than 0.6. So the sea salt cannot be picked. According to above two steps, low DEP and high CR corresponds to polluted dust, not sea salt.

7. L240, if dust is reported at stations and dust layer is detected above PBL by lidar, is it pure or transported?

We thought it was pure dust. Because pure dust is accompanies with dust days, like dust storm, blowing dust and floating dust. For every case we picked out, IDL procedure was fist conducted, and then we confirmed them with our eyes. Under your constructive suggestion, we have modified our detection method and picked pure dust and polluted dust again.

8. L318, what is the target of statistical analysis? All data during October 2009 and June 2013? Or, some restriction by scattering ratio? What is the height range?

We did statistical analysis to find the threshold between pure dust and polluted dust from the optical perspective and further to improve the detection of different aerosol type in numerical modeling and satellite algorithm. Not all data during October 2009 and June 2013. Owing to the output energy of the NIES equipment low, some data during this period is not useful. Also we have conducted strict data quality control. Every case was strictly picked out using IDL source code and after that we confirmed every case personally with our eyes. The original data height range is 0 to 18 km, but considering our requirements, 0 to 6 km above the ground is chosen because aerosols are concentrated in this height range.

9. L367, what is the physical meaning of skewness and kurtosis for histograms?

Mean value alone cannot describe the distribution of pure dust and polluted dust, so we add skewness and kurtosis to help us to detect them clearly in the space-born lidar and numerical modeling. Skewness is a measure of the direction and extent of skewness in the distribution of statistical data and is a numerical feature of the degree of asymmetry in the distribution of statistical data. The number of features characterizing the degree of asymmetry of the probability distribution density curve with respect to the mean. Intuitively, it is the relative length of the tail of the density function curve. In our results, take depolarization ratio for example, skewness of pure dust and polluted dust are greater than 0, which means they all located on the right less than the left. But the skewness of pure dust is smaller than that of polluted dust which means for pure dust

the number of large values is larger. So using this distribution we can constrain the satellite observation and model results when detecting dust aerosols.

10. Figure 1, describe the time period in which the number of dust events were counted.

The time period in Figure 1 is 2013, one year data.

11. Figure 3 and 5, unit for panel (a) is unnatural. Is it 10Ë̈E̦-2/km/sr?

Thank you for your kind reminding. We have corrected in the corresponding position.

12. Figure 3, PBL height at 0 UTC was above the cloud layer. How lidar can detect it without effective signal?

Thank you for your kind reminding. As you said, when it is cloudy, we cannot get accurate PBL. The case in Figure 3 is not appropriate and we modified the selection method and changed another case.

13. Figure 4 and 6, all trajectories touch the ground. Are these paths reliable?

Thank you for your kind reminding. Out of our ignorance, those paths may not be reliable. According to the path and altitude, we picked out every case again.

14. Figure 9, (b) for pure and (a) for anthropogenic dust.

Thank you for your kind reminding. We have corrected in the corresponding position.

Technical Corrections:

1. L53, L58, L89 etc, unify the usage of "," and ";".

Thank you for your kind reminding. We have corrected in the corresponding position.

2. References, J. Quant. Spectrosc. Radiat. Transfer

Thank you for your kind reminding. We have corrected in the corresponding position.

---

## Author Comment (AC6) · 10 Mar 2018

General Comments:

Unfortunately, the paper is unacceptable. The location of the lidar observations (SACOL site) is excellent. The lidar data set is probably of high quality. So I would like to encourage the authors to resubmit the paper after considering my suggestions. The main reason for rejection is that the authors fail to provide a clear definition and thus separation of pure dust and anthropogenic dust cases. A clear definition can be

done by means of the particle linear depolarization ratio. But the authors only present volume depolarization ratios. These values vary with the relative amount of dust, and thus can be low even in the case of pure dust, and large, even in the case of polluted dust. So the only way is: Compute the particle depolarization ratio and use this parameter to distinguish polluted (or anthropogenic) and pure dust cases. If the particle depolarization ratio is > 25% one may call the event a pure dust case and if we have <25% then we may call it a polluted dust case. Furthermore, most of the results are simply given in terms of attenuated backscatter. This quantity varies with the amount of aerosol, so with the amount of dust and/or pollution. We need the particle backscatter coefficient to describe aerosol properties with height. The overall impression is: The paper is to 80% just based on 'opinions', and not on 'objective' facts. The lidar community dealing with dust research would be upset if this low-quality paper gets published in its present form. The authors may want to resubmit their paper. Then the analysis must be fully based on (a) particle backscatter coefficients for 532 and 1064 nm, (and not on 532 nm attenuated backscatter) and (b) on particle depolarization ratios (and not on volume depolarization ratios). The particle depolarization ratio can be easily computed from the volume depolarization ratio and the 532 nm particle backscatter coefficient (see the cited publication of Freudenthaler 2009, or some papers from the NIES group). And then introduce a clear criterion for anthropogenic dust, based on the particle linear depolarization ratio.

Thank you for your serious review. First, we just used the data from SACOL and not involved in the observation and inversion process. Second, linear volume depolarization ratio is provided by SACOL group and is used as a part to detect pure dust and anthropogenic dust. This method is the same with Huang et al. (2015) and Liu et al. (2005). This work is mainly to further prove the detection method of anthropogenic dust in Huang et al. (2015). On the other hand, we all think using the particle depolarization ratio and particle attenuated backscatter coefficient is more accurate when detecting pure dust and polluted dust. And after we received your suggestion, we were always trying to calculate particle backscatter coefficients for 532 and 1064 nm and the particle linear depolarization ratio following the method in the literature of Fredenthaler et al. (2009). But time is not enough, we have not obtained satisfied results. And we will insist with improving our detection method and get more accurate results in the future research.